# Integrated Metabolomics and Transcriptomics Using an Optimised Dual Extraction Process to Study Human Brain Cancer Cells and Tissues

**DOI:** 10.3390/metabo11040240

**Published:** 2021-04-14

**Authors:** Alison Woodward, Alina Pandele, Salah Abdelrazig, Catherine A. Ortori, Iqbal Khan, Marcos Castellanos Uribe, Sean May, David A. Barrett, Richard G. Grundy, Dong-Hyun Kim, Ruman Rahman

**Affiliations:** 1Children’s Brain Tumour Research Centre, School of Medicine, Biodiscovery Institute, University of Nottingham, Nottingham NG7 2RD, UK; mszaw2@exmail.nottingham.ac.uk (A.W.); msxap16@exmail.nottingham.ac.uk (A.P.); mgzrgg@exmail.nottingham.ac.uk (R.G.G.); 2Centre for Analytical Bioscience, Advanced Materials and Healthcare Division, School of Pharmacy, University of Nottingham, Nottingham NG7 2RD, UK; pazsma@exmail.nottingham.ac.uk (S.A.); paaco1@exmail.nottingham.ac.uk (C.A.O.); paadb1@exmail.nottingham.ac.uk (D.A.B.); 3National Arabidopsis Stock Centre, Plant and Crop Sciences, School of Biosciences, University of Nottingham, Nottingham NG7 2RD, UK; sbziak@exmail.nottingham.ac.uk (I.K.); sbzmc3@exmail.nottingham.ac.uk (M.C.U.); sbzstm@exmail.nottingham.ac.uk (S.M.)

**Keywords:** dual-extraction, cancer, metabolite, RNA, integrated omics

## Abstract

The integration of untargeted metabolomics and transcriptomics from the same population of cells or tissue enhances the confidence in the identified metabolic pathways and understanding of the enzyme–metabolite relationship. Here, we optimised a simultaneous extraction method of metabolites/lipids and RNA from ependymoma cells (BXD-1425). Relative to established RNA (mirVana kit) or metabolite (sequential solvent addition and shaking) single extraction methods, four dual-extraction techniques were evaluated and compared (methanol:water:chloroform ratios): cryomill/mirVana (1:1:2); cryomill-wash/Econospin (5:1:2); rotation/phenol-chloroform (9:10:1); Sequential/mirVana (1:1:3). All methods extracted the same metabolites, yet rotation/phenol-chloroform did not extract lipids. Cryomill/mirVana and sequential/mirVana recovered the highest amounts of RNA, at 70 and 68% of that recovered with mirVana kit alone. sequential/mirVana, involving RNA extraction from the interphase of our established sequential solvent addition and shaking metabolomics-lipidomics extraction method, was the most efficient approach overall. Sequential/mirVana was applied to study a) the biological effect caused by acute serum starvation in BXD-1425 cells and b) primary ependymoma tumour tissue. We found (a) 64 differentially abundant metabolites and 28 differentially expressed metabolic genes, discovering four gene-metabolite interactions, and (b) all metabolites and 62% lipids were above the limit of detection, and RNA yield was sufficient for transcriptomics, in just 10 mg of tissue.

## 1. Introduction

Integrated omics techniques can improve the understanding of mechanisms through which a cellular phenotype is generated, where each technique allows insight into cell regulation and function at a different but inter-related level of the genome [1,2]. The metabolome, although dependent on the genome, is in constant flux within a changing biological environment, and provides an acute, dynamic molecular snapshot of the observed phenotype of a disease [3]. The transcriptome reveals which genes are expressed/repressed and contribute to the phenotype, allowing inference from the genome landscape (which could be aberrant due to DNA mutations or chromosomal rearrangements) [4]. If the expressed genes encode metabolic enzymes, then such expression levels could directly help to understand an increase in activity of a metabolic pathway and aid in the discovery of drug targets for diseased cells, amenable to pharmacological inhibition [5,6]. Similarly, aberrantly expressed genes which encode non-enzymatic proteins may also be reflective of aberrant metabolism within a diseased cell or sub-cellular niche, albeit indirectly [7].

Combining metabolomics and transcriptomics data from the same cells or tissues would allow a much better holistic understanding of how the cell generates the aberrant metabolome of a disease such as cancer, how the cell compensates for variability and evolutionary dynamics within the tumour microenvironment, and how the metabolome feeds back signals to the cell [3,8,9]. Accurate observation of the dynamic interaction between transcribed gene products and metabolites would allow a more systems biology read-out in disease models [3,10]. For example, Heiland et al. discovered that they could separate glioblastoma into four sub-types based on combined analysis of the metabolome and transcriptome. For instance, Cluster 2 was defined as a mesenchymal/immune-response-related cluster after enrichment of mesenchymal and immune response genes, and exclusive enrichment of tryptophan, phosphocholine and choline metabolism (which are known to play a role in immune responses) [9]. Kucharzewska et al. reported that under hypoxic conditions in vitro, increased transcription of glucose uptake and glycolysis genes was associated with higher levels of glucose and glycolysis intermediates in glioblastoma cells [11].

Published methods typically extract RNA and metabolites from different regions of a given tissue specimen, which limits integration of datasets as neither the cell genotype nor phenotype is necessarily homogenous throughout tissue, particularly in malignant cancerous tissue [9,12,13]. Single-cell transcriptomics has shown that there is even cell-to-cell heterogeneity [14]. The relationship between RNA transcript level and metabolite level is non-linear; therefore, to deduce any relationship, the biological variables must be minimised by using exactly the same sample handling process for both metabolomics and transcriptomics, known as a dual-extraction process [3,8,15]. An additional limitation of prior methodologies is the preclusion of integrated omics due to scarcity of available tissue, where this is insufficient for two distinct methods of extraction [13,15]. To overcome these limitations, dual extraction methods enable multi-omics from the same population of cells or tissue fragment [13,16], which potentially allows a more physiologically accurate integration of omics datasets.

Three simultaneous RNA/metabolite extraction methods exist, optimised for plant tissue, microbes and human cell lines [15,16,17]. Recently, a new method has emerged, in which the non-polar metabolite (lipid) extraction is optimised for oxylipins [18]. The three methods used different ratios of methanol, chloroform and water, and different physical extraction methods varying from cryomilling to rotating or shaking the sample, to extract the metabolites. RNA extraction could be either by phenol-chloroform extraction, solid phase extraction, or both. RNA extraction always follows the metabolite extraction step in these methods, and therefore it is possible that RNA is lost in the metabolite extraction step. The metabolite extraction uses lipophilic solvent to open the cell membrane [15] and polar solvents which could dissolve and extract the RNA, though most RNA is expected to precipitate and form a pellet rather than dissolve [10]. Cryomilling skin tissue after storage in RNA later or after flash freezing led to RNA degradation, while cryosectioning after snap freezing preserved the RNA [19]. In 36/42 samples of various tissues, from which a low 28S/18S ratio was observed despite high RNA integrity number, the cryomill was used rather than cryosectioning; these data suggest that a cryomill has the potential to mechanically shear transcripts and oligomerise them by producing heat [19]. It is therefore important to assess whether the RNA yield is reduced by a dual extraction method compared to extracting the RNA first. For some samples, the RNA degrades quickly and must be stabilised [19], precluding prior metabolite extraction; for example, RNA from colonic tissues must be pre-treated with an RNA stabilisation solution [20], while polyphenols and secondary metabolites affect the recovery and purity of RNA in plants [15,21]. The integrity and recovery yield of nucleic acids can be increased by using silica columns, compared to other methods [15]. Using acetonitrile was better than using ethanol for RNA purification, as it removes proteins from the RNA [15]. However, the metabolite extraction step might aid RNA extraction as it could purify the samples by removing most of the compounds that interfere with commercial RNA extraction buffers [15]. A limitation of these three methods is that they provide little explanation of what the detected molecules are (explaining that the method gave a total ion chromatogram with clear numerous well-defined peaks [17], or that methanol/chloroform gives sharp chromatograms and facilitates membrane disaggregation due to the solubilisation of lipids [15] while a distinct previous method from the same research team detected and quantified 652 metabolites over a wide range of classes [10]), or they produce targeted assays of just a few detected metabolites [16] with intensities 5–15 times higher than that previous method [10]. To obtain good-quality omics data, the molecule concentrations must be greater than the limits of instrument detection. Moreover, to enable integration, the coverage must be complete and untargeted to avoid bias/overfitting of a supervised model to a small number of significant features/metabolites. The physicochemical properties of metabolites affect their extraction efficiencies by any one particular method, meaning that each extraction protocol has a systematic bias for extracting molecules of a certain polarity from cells, disabling complete coverage [16].

A recently published method used an RNA lysis buffer to extract metabolites, and the cryomilled lysate was split into equal sized fractions for metabolomics and transcriptomics, which prioritises RNA extraction and quality but is not an RNA-first method, and does not overcome the problem of limited amounts of tissue, and so is not a simultaneous extraction method [13]. The aim of this work was to develop a robust dual extraction method for metabolites and RNA that is applicable to brain cancer cells and tissues. Although there is one existing dual protocol for human cells (Jurkat T cells) [16], this was not optimised for the classes of metabolites that were of interest to us since it focused on central carbon and nitrogen metabolism, whereas brain-specific molecules such as homocarnosine, N-acetylaspartate, gamma-aminobutyric acid, and lipids were known to be important in brain tumours [11,22,23,24,25,26]. In particular, a large volume of the non-aqueous, less polar solvent is needed to fully extract the lipids from brain tissue [27]. Two existing single extraction and four dual extraction protocols were critically assessed based on their practical reproducibility, the suitability of the metabolite extraction protocol to human cancer cells and their RNA recovery. The effectiveness of the optimised method was verified by testing the sensitivity to a) discriminate both subtle and profound alterations in transcription and metabolism in an acute serum starvation period in a human ependymoma cell line; and b) detect metabolites and RNA in increasingly smaller masses of human brain tumour tissue.

## 2. Results and Discussion

### 2.1. Assessment of the Dual Extraction Methods for Metabolite/Lipid Extraction from Human Brain Cancer Cells

The dual extraction methods of metabolites/RNA (*n* = 2–4; methanol:water:chloroform ratios are shown in brackets): cryomill/mirVana method (1:1:2) [17], cryomill and wash/Econospin columns method (5:1:2; herein referred to as cryomill-wash/Econospin) [15], cell extract rotation/phenol-chloroform method (9:10:1; herein referred to as rotation/phenol-chloroform) [16] and sequential solvent addition and shaking/mirVana method (1:1:3; herein referred to as sequential/mirVana), as detailed in the methods/supplementary methods, were compared to a standard routine method for metabolite extraction [28], which was used as a positive control (metabolomics reference method)**,** sequential solvent addition and shaking (1:1:3). The total ion chromatogram of the extracts in both zwitterionic-polymer–hydrophilic interaction liquid chromatography-liquid chromatography-mass spectrometry (ZIC-pHILIC-LC-MS, metabolites) and reverse phase-liquid chromatography-tandem mass spectrometry (RP-LC-MS/MS, lipids) analyses showed adequate separation and smooth peaks (Appendix A). The total ion counts (TICs) of the quality controls (QCs) had a relative standard deviation (RSD) of 7% and 6% in metabolomics and lipidomics analyses, respectively, which was deemed an acceptable level of variation by Hutschenreuther et al. and Fiehn et al. [29,30,31]. Additionally, the analytical performance of the LC-MS was assessed using the pooled QC approach using Principal Component Analysis (PCA) [17]. The QCs were clustered together in the centre of the PCA score plots (Appendix A), indicating that the LC-MS analytical performance was satisfactory for untargeted metabolomics.

#### 2.1.1. Metabolite Analysis

Cryomill-wash/Econospin produced the highest total ion count (versus the sequential solvent addition and shaking reference method, *p* < 0.05), which may have been due to the washing step employed, ensuring all water-soluble metabolites were concentrated in the aqueous phase free from lipids (Table 1) [32]. The total number of the identified metabolites in BXD-1425 cells was 193, across all extraction methods. Therefore, the methods did not discriminate certain classes of metabolites; for instance, non-essential amino acids, except for cysteine, were detected and identified with all methods. Additionally, B vitamins, which have essential roles in the cell metabolism, and four tri-carboxylic acid (TCA) cycle intermediates were detected. PCA showed that there was no separation or clustering between groups and all groups overlapped, concluding that all the methods were equally capable of extracting metabolites from BXD-1425 cells grown in vitro (Appendix A).

Cryomill/mirVana significantly increased the extraction efficiency of 22 metabolites and yet decreased the extraction efficiency of a further nine compared to sequential solvent addition and shaking (Table 1), while cryomill-wash/Econospin and sequential/mirVana did not change the extraction efficiency of any metabolite significantly, and rotation/phenol-chloroform decreased the amount of just one metabolite. Therefore, cryomill-wash/Econospin, rotation/phenol-chloroform and sequential/mirVana were equally as good as our standard metabolomics reference method, sequential solvent addition and shaking. From a metabolomics perspective, although cryomill/mirVana increased some metabolite concentrations, the procedure was difficult to practically perform since the cells needed to be cryomilled whilst in the presence of the solvent, and that caused one of the tubes to break and the solvent to leak. This is an added risk when working with irreplaceable samples. Therefore, the method most suitable for simultaneous lipids and RNA extraction was investigated as an outcome measure, since any of Methods C–E could be used for polar metabolites extraction.

#### 2.1.2. Lipids Analysis

The low fraction of chloroform used in rotation/phenol-chloroform meant that the extraction was monophasic and therefore no lipidomics was performed. PCA showed that the extracted lipid profiles using sequential/mirVana were similar to the metabolomics reference method (sequential solvent addition and shaking), whereas cryomill/mirVana and cryomill-wash/Econospin lipid profiles were very different (Appendix A) as expected, because sequential solvent addition and shaking and sequential/mirVana are equivalent. More lipids were extracted using cryomill-wash/Econospin compared to cryomill/mirVana and sequential/mirVana (Table 2); initially a low percentage of chloroform was used, but the phase separation likely concentrated the water-insoluble lipids into the chloroform phase and separated the layers more easily so that the majority of the chloroform could be collected. Additionally, the number of lipids with differential abundance was highest with cryomill-wash/Econospin, with 64% of those increasing in abundance. Cryomill/mirVana and sequential/mirVana were marginally better than sequential solvent addition and shaking for the number of lipids extracted and both had similar numbers of lipids which increased and decreased in abundance.

### 2.2. Assessment of the Dual Extraction Methods for RNA Extraction from Human Brain Cancer Cells

All the extraction methods were compared to a standard method for RNA extraction using the mirVana^TM^ miRNA Isolation Kit (Ambion, Life Technologies, Carlsbad, CA, USA) (Transcriptomics Reference Method). All the extraction methods produced RNA of sufficiently high purity (A_260_/A_280_ ratios of approximately 2.0) and quality for downstream applications such as transcriptomic microarrays and RNA sequencing, since Roume et al. recommended an RNA integrity number (RIN) threshold of 7.0 [16,17], which all samples surpassed (Table 3). The amount of RNA obtained from a total of 2 million cells was ≥ 20.0 µg by all methods, sufficient for transcriptomics analysis which typically requires ≥ 0.5 µg of RNA [16]. However, since it may be necessary to use a lower number of cells in certain cases, a recovery % was carried out in comparison to the positive control (Transcriptomics Reference Method, MirVana kit) and at least 30% of the RNA was lost during the metabolite extraction step. The lowest amount of RNA was recovered with rotation/phenol-chloroform, which may be due to the lack of a solid phase extraction step to isolate RNA. Another potential reason for the lower recovery of RNA by rotation/phenol-chloroform and cryomill-wash/Econospin could be that using less chloroform reduced the precipitation of RNA. Therefore, cryomill/mirVana and sequential/mirVana have been identified as optimal, as they both use the mirVana^TM^ miRNA isolation kit similar to the positive control that may account for the better recovery observed.

In conclusion, cryomill/mirVana was able to extract many metabolites with higher abundance, and cryomill/mirVana and sequential/mirVana recovered both high-quality RNA and a high RNA yield compared to the positive control and enabled both metabolomics and lipidomics analysis for a greater coverage of the metabolome.

### 2.3. The Effects of Acute Cellular Stress on Metabolism and Transcription in BXD-1425 Cells

As a proof-of-concept, the selected dual extraction sequential/mirVana method was then applied to study the difference in metabolism and transcription induced by acute cellular stress, specifically by starving cells of serum. Serum starvation was chosen, as deprivation of cellular nutrients leads to an abrupt growth arrest in mammalian cells, characterised initially by RNA expression changes and metabolite level changes. Thus, comparing metabolomics and transcriptomics between serum-deplete versus serum-replete brain cancer cells offers a rapid means to verify this dual extraction method on a genome-wide level.

Multivariate analysis using PCA and OPLS-DA showed that serum-starved and serum-replete samples were statistically discriminated (Appendix A, Figure 1). PCA showed that intra-group variability was much less than inter-group variability and 82.5% and 90.2% of the variance in the metabolome and lipidome, respectively, was explained by PC1 and PC2. The OPLS-DA models were cross-validated and the degree of fit of the OPLS-DA of the metabolomics dataset was high (R^2^X 0.814, R^2^Y 0.995) and the predictive ability (Q^2^) of the model was much greater than 0.40 (the threshold for a biological model [33]) and approached the theoretical maximum of 1.00 (Q^2^ = 0.979) [33,34]. Therefore, the model is validated and the degree of overfitting is minimal. Sixty-four metabolites were sufficient to distinguish serum-starved from serum-replete cells (Figure 2, Appendix A). Many metabolites were relatively less abundant in the serum-starved cells including TCA cycle compounds (succinate and malate), amino acids (L-alanine, L-aspartate, L-glutamate, L-methionine, L-serine, L-phenylalanine and L-tyrosine), molecules which shuttle fats into the TCA cycle (O-acetyl carnitine and carnitine), and antioxidants (ascorbate and glutathione), despite increased levels of pyruvate and L-glutamine, suggesting cells were in the early stages of apoptosis [35,36,37]. The degree of fit of the OPLS-DA model of the lipidomics dataset was also high (R^2^X 0.880, R^2^Y 0.998) with Q^2^ = 0.994; therefore, the model is validated, and degree of overfitting is minimal. Ninety-seven lipids significantly altered in relative abundance could discriminate the treatment from the control cells (Appendix A). The classes of these lipids, in order from most to least common (number of lipid species shown in brackets), were triglyceride (31), phosphatidylcholine (29), ceramides (13), phosphatidylethanolamine (12), sphingomyelin (5), lysophosphatidylcholine (2), diglyceride (2), simple Glc series (1), phosphatidylinositol (1) and cholesterol ester (1). Notably, PI(18:0_20:4) (2.4 fold change), the most common phosphatidylinositol in cell membranes, and cholesterol ester (18:2) (0.1 fold change), the most common cholesterol ester in plasma and third most common in liver tissue, were differentially abundant in serum-starved versus serum-replete cells [38,39]. In the ceramides class, the following lipids had a fold change of less than 0.5: Cer(d18:1_18:0), Cer(d18:1_20:0), Cer(d18:1_23:0), Cer(d18:2_23:0) and Cer(d16:1_23:0). This contrasted with the phosphatidylethanolamines, which had a fold change of >2: PE(18:2e_16:0), PE(18:0p_18:1), PE(16:0p_20:4) and PE(16:0p_22:5). Ceramides can activate in the endoplasmic reticulum and mitochondria, the major pathways regulating cell death, and previous work found that ceramide levels increase during apoptosis [40]. This contrasts with our result, as we found lower levels of ceramides and of sphingomyelins in serum-starved cells. Those phosphatidylcholines, phosphatidylethanolamines and phosphatidylinositol putatively containing the fatty acid arachidonate (20:4(ω-6)) were increased in abundance under serum starvation whilst the genes CYP1B1 and PTGDS which encode for enzymes of arachidonic acid metabolism and prostaglandin formation were increased in expression (*vide infra*), and as leukotrienes and prostaglandins are signalling molecules, this indicated increased paracrine signalling in the absence of serum [41] (pp. 375, 824). The differentially abundant metabolites and lipids detected in this experiment are extracted directly from cells (metabolic fingerprint) and showed the developed dual extraction method was able to detect intracellular metabolic changes in the BXD-1425 cells upon an acute period of serum starvation, and it is likely to be suitable for the detection of other metabolic alterations, such as those associated when treating cancer cell lines with chemotherapeutic drugs or depriving them of lipoproteins [42,43].

Transcriptomic array data were analysed and compared using PCA, which showed that serum-starved and control samples were statistically discriminated and explained 65.8% of the variance in gene expression, with PC1, PC2 and PC3 accounting for 33.8%, 18.1% and 13.9% of the variance, respectively (Figure 3a). Filtering was performed to remove genes where expression did not differ from that in controls and genes where hybridization signals fell below the threshold for reliable detection using Partek^®^ Genomics Suite^®^. To reduce false positives, a Benjamini–Hochberg multiple correction test was applied to generate a final list of genes consisting of 128 transcripts (Figure 3b), with 28 genes mapping onto metabolic pathways (Table 4). A total of 71 genes were upregulated in the serum-starved cells, including many angiogenesis and cell adhesion genes such as SAT1 and MFAP4. Fifty-seven genes were downregulated compared to the serum-replete cells, comprising cell morphogenesis and DNA repair genes such as GREM1 and UBE2T. One of the biggest functional differences between samples consisted of the significant enrichment of immune response genes such as KLRC3 and CXCL6 in serum-starved cells compared to the serum-replete group.

### 2.4. Multi-Omics Integration

Cytoscape software with MetScape plugin was used to examine the potential pathway-based associations between the two omics datasets [2]. Data integration revealed interactions occurring on 37 metabolic pathways in serum-starved relative to serum-replete BXD-1425 cells, including 1525 nodes (entities of interest) and 6161 edges (associations between entities) with the largest number of associations occurring on the purine metabolism pathway (Figure 4 and Appendix A). Four nodes presented with direct statistically significant gene-metabolite interactions with 3/4 mapping onto the arginine, proline, glutamate, aspartate and asparagine metabolism pathways (Figure 5). For example, an increase in concentration of L-proline and increase in expression of P4HA1 (prolyl 4-hydroxylase subunit alpha 1), which catalyses the transformation of L-proline into 4-hydroxyproline, in serum-starved cells (Figure 5b; 4-hydroxyproline not detected), suggests the shuttling of L-proline to pyruvate to enter the TCA cycle under serum starvation. Therefore, dual extraction of metabolites and RNA from the same cells enabled accurate integration of metabolomics and transcriptomics data, and the expression levels of genes encoding metabolic enzymes have aided in understanding the genetics underlying an increase in activity of a metabolic pathway.

Understanding cell metabolism in the absence of signals from serum is somewhat a mimic for the poorly vascularised, nutrient-, growth factor- and oxygen-deficient core of tumours, especially when combined with lowered glucose media and growth in a hypoxia chamber [44], and therefore hints at why brain cancer survives. Cancer cells are dependent on antiapoptotic pathways and therefore cellular stress including growth factor deprivation, DNA damage, and treatment with most anticancer drugs will activate the intrinsic pathway of apoptosis [45]. Evidence of the early stage of apoptosis from our data suggests pathways which can be targeted with drugs that will induce apoptosis, a promising concept [45]. For example, it has been discovered that growth signals lead to proteins being phosphorylated on tyrosine residues to which PKM2 binds, which negatively regulates PKM2 activity [45]; therefore, the removal of serum growth signals may have led to higher PKM2 activity, causing the higher level of pyruvate observed under serum-starvation. Using an allosteric activator of PKM2 such as fructose 1,6-bisphosphate would potentially trigger apoptosis in cancer cells.

### 2.5. Application of Sequential/mirVana Method to Primary Brain Tumour Tissue

Though the optimal method is suitable for in vitro cultures of brain tumour cells, we sought to determine whether it would be suitable for primary brain tumour tissue and how much tissue would be needed. Therefore, we used the same method, with homogenisation of the tissue into cells, on masses of 10 to 40 mg of ependymoma tissue from one patient (15/243), extracting metabolites, lipids and RNA. The metabolites and lipids were analysed by LC-MS and LC-MS/MS, and the RNA was measured by Nanodrop.

All metabolites identified in the tumour of patient 15/243 were detected in 10, 20 and 40 mg samples (Appendix A). The general trend was for the amount detected to increase with increasing mass of tumour; however, each metabolite responded to the detector differently. Of the 39 metabolites which were matched to standards (level 2 identification), eight exhibit a linear response to the detector.

The number of lipid groups detected in the tumour (lipids grouped according to their summed fatty acids), increased as the mass of tissue increased. Two hundred lipid groups were extracted from 10 mg of tissue, 258 were extracted from 20 mg and 325 from 40 mg of tissue (Appendix A). Peak areas of lipids in a certain class were combined and the total peak areas of four of the six most common lipid classes in the samples showed saturation at the detector when 40 mg of tissue was used (that is, there was a deviation from linearity). These classes were ceramides, phosphatidylcholines, phosphatidylethanolamines and sphingomyelins, and did not include diglycerides and triglycerides (Appendix A, Appendix A).

The RNA mass extracted from tissue roughly doubled from 10 (7.6 µg RNA) to 20 (14.7 µg) to 40 mg (36.5 µg) tissue, which suggests that the yield of RNA is proportional to the amount of tissue used. The purity of the RNA was measured as 260/280 and 260/230 ratios: 10 (1.99, 2.07), 20 (2.01, 2.08), and 40 mg (2.03, 1.70). The yield and purity of RNA were suitable for consideration of transcriptomics analyses including RNA sequencing, from a minimum of 10 mg of tissue.

## 3. Materials and Methods

Accession numbers have not yet been obtained—they will be provided before publication.

### 3.1. Preparation of Cell Pellets

BXD-1425 cells were selected as a model in this study due to their in vitro stability and rapid growth rate [46]. BXD-1425 cells were derived from a paediatric patient with a recurrent ependymoma, World Health Organisation Grade III tumour. A total of 2.0 × 10^6^ of BXD-1425 cells were detached using 1×Trypsin-EDTA, counted from a sub-confluent, log phase T175 flask and centrifuged to remove the medium. The cells were then re-suspended in PBS (1 mL in a 2 mL tube) and centrifuged at 305 rcf for 5 min at room temperature. The supernatant was removed and the remaining cell pellets were flash-frozen in liquid nitrogen and stored at −80 °C for the analysis.

### 3.2. Single Extraction of Metabolites or RNA

#### 3.2.1. mirVana Kit

*mirVana kit:* Extraction of RNA was performed according to the manufacturer’s protocol for the mirVana^TM^ miRNA Isolation Kit (Ambion, Life Technologies, Carlsbad, CA, USA), with 500 µL lysis/binding solution being used and the RNA being eluted in 50 µL of nuclease-free water (Transcriptomics Reference Method, a standard RNA kit). The extracted RNA was stored at −20 °C.

#### 3.2.2. Sequential Solvent Addition and Shaking

Extraction of metabolites was performed using a Metabolomic Reference Method for standard metabolites extraction [28]. Cold methanol (200 µL, −80 °C) was added to the cell pellet just thawed on ice, and the content was shaken at 300 rpm at 4 °C for 30 min. Chloroform (600 µL, 4 °C) was added, vortexed gently and left on ice for 10 min; the procedure was repeated three times. Water (200 µL, 4 °C) was added to the extract, vortexed gently and centrifuged at 20,000 rcf for 10 min at 4 °C. The upper aqueous and lower organic phases of the extracts were separated.

### 3.3. Dual Extraction of Metabolites and RNA

Four different dual extraction methods of metabolites and RNA were investigated and designated as: cryomill/mirVana [17], cryomill-wash/Econospin [15], rotation/phenol-chloroform [16] and sequential/mirVana (Figure 6). The methods used from the literature are written in full in the Appendix A.

#### 3.3.1. Cryomill/mirVana: Cryomill and mirVana kit (Phenol-Chloroform and SPE)

The procedure was followed as stated elsewhere, including Section 3.2, Section 3.3, and a variant of Section 4 that used the mirVana^TM^ miRNA Isolation Kit as outlined in mirVana kit without cryomilling [17]. Note that a Mixer Mill 301 (Retsch, Haan, Germany) was used.

#### 3.3.2. Cryomill-wash/Econospin: Cryomill, Wash and Econospin Columns (SPE)

The procedure was followed as outlined by Valledor et al. using the Mixer Mill 301 at 25 Hz with cooled autoclaved-sterilized milling balls (5 × 2 mm + 2 × 5 mm), and a thermal shaker for the RNA pellet at 300 rpm for 30 min [15].

#### 3.3.3. Rotation/Phenol-Chloroform: Rotate Cell Extracts and Phenol-Chloroform

The protocol *C* described by Vorreiter et al. was followed starting from resuspension in PBS [16]. The RNA pellet was re-suspended in 15 µL of nuclease-free water (60 °C).

#### 3.3.4. Sequential/mirVana: Sequential Solvent Addition, Shake and mirVana Kit (Phenol-Chloroform and SPE)

The extraction procedure started with using sequential solvent addition and shaking for the extraction of metabolites, followed by the extraction of RNA using the mirVana kit from the solid interphase between the aqueous and the organic phases.

### 3.4. Serum Starvation

BXD-1425 cells (human ependymoma cell line) were grown in DMEM with 10% FBS in T75 flasks until 70–80% confluent. A complete media change was made to either DMEM (serum-starved treatment) or DMEM with 10% FBS (serum-replete control), with four replicates produced. An experimental period of 28 h was chosen based on a previous study, where although an increase in VEGF-promotor activity from 3–24 h post-serum starvation was detected by a promoter-reporter construct, it was not until after 24 h that an increase in VEGF mRNA expression was noticeable by Northern blot [47], whereas a period that is too long will lead to cell death. After 28 h, the medium was removed, and the cells were washed with 1 × PBS (5 mL, 37 °C) and quenched by adding methanol (0.7 mL, −80 °C). The flask was placed on ice and the cells were scraped from the flask surface and transferred with the methanol into an ice-cold 1.5 mL tube and stored on dry ice. Cells from two further replicate flasks were counted using a haemocytometer. Tubes were shaken at 2000 rpm for 30 min at 4 °C. Extracts were transferred to 15 mL tubes and chloroform added (2.1 mL, 4 °C). The tube was vortexed gently and left on ice for 10 min and this procedure repeated three times. Water (0.7 mL, 4 °C) was added and the tube was vortexed gently and centrifuged for 10 min at 4000 rcf at 4 °C. The upper and lower phases of the extracts were separated. The interphase was stored at −80 °C and later processed using the mirVana^TM^ miRNA Isolation Kit, using 500 µL of lysis/binding buffer. The mixture was transferred to an Eppendorf tube after adding the homogenate additive and leaving on ice. Total RNA was eluted into 2 × 50 µL of RNase free water and stored at −80 °C.

### 3.5. LC-MS Sample Preparation

In all methods, the aqueous and the organic phases were separated, the solvents were evaporated using the Jouan Centrifugal Evaporator (Thermo Fisher Scientific, Hemel Hempstead, UK) at room temperature, and the extract residues were stored at −80 °C. For LC-MS, the aqueous and the organic extracts were reconstituted in 100 µL of methanol (4 °C) and 100 µL of isopropanol (4 °C), respectively, and centrifuged at 13,000 rcf, 4 °C for 10 min. Then, 70 µL from each was transferred into HPLC vials and stored at −80 °C until analysis; in addition, 20 µL from each sample were mixed as a pooled quality control (QC).

### 3.6. Applying Sequential/mirVana Method to Extract Metabolites, Lipids and RNA from Brain Tumour Tissue

The study utilised tissue derived from a paediatric brain tumour patient treated at the Queen’s Medical Centre, Nottingham. The study was approved by the National Research Ethics Committee (NRES Committee East Midlands) (Reference Number 11/EM/0076). Tissue samples were cut from the ependymoma tumour of patient 15/243 in a sterile Petri dish over dry ice. Samples were homogenised using a Stuart Homogeniser SHM1 1116 in methanol (100 µL, 4 °C) inside 2 mL tubes over dry ice. Chloroform (300 µL, 4 °C) was added and vortexed well. Water (100 µL, 4 °C) was added and vortexed well. Samples were centrifuged at 13,000 rcf for 10 min at 4 °C. Parts of the upper phase (100 μL) and lower phase (200 μL) were collected into 1.5 mL tubes. The upper phase was centrifuged (as before but for 5 min) and the solvent transferred to a HPLC vial. The lower phase solvent was evaporated using the Jouan Centrifugal Evaporator at room temperature (25 min) and the residue was resuspended in isopropanol (100 µL, 4 °C), then centrifuged (as before) and the solvent transferred to a HPLC vial. The 2 mL sample tube was centrifuged again (as before) and the remaining solvent removed carefully to leave the solid interphase. RNA was extracted from the interphase using the mirVana^TM^ miRNA Isolation Kit, using 500 μL lysis buffer and eluting into 2 × 50 μL of RNase free water.

### 3.7. LC-MS Analysis

Experimental samples were randomised through the sequence. In addition, the pooled QC was run after every six samples to assess the instrument performance. Solvent blanks were run prior and subsequent to samples.

Metabolites were separated using an Accela UHPLC system coupled to an Exactive Orbital trap MS (Thermo Fisher Scientific, Hemel Hempstead, UK) as previously described [48,49]. Metabolite extracts were injected (10 µL, 4 °C) onto a ZIC-pHILIC column (150 × 4.6 mm; 5 μm particle size; Merck SeQuant, Darmstadt, Germany) held at 45 °C. The starting mobile phase was 20% mobile phase A (20 mM ammonium carbonate in water, pH 9.1) and 80% mobile phase B (100% acetonitrile). The constant flow rate was 300 µL/min. The proportion of mobile phase A increased up to 95% from 0 to 15 min (linear gradient) and then decreased back to 20% in 2 min, where it was held for 7 min for re-equilibration, the total elution time was 24 min. Different mixtures of 268 authentic standards were analysed with the samples for metabolite identification. Mass spectra of eluted extracts were acquired using the Exactive MS fitted with a heated ESI (HESI) source. Full MS profiling with a simultaneous ESI+ and ESI- switching were used with *m/z* 70–1400 range at 50,000 resolution. The probe and capillary temperature were maintained at 150 and 275 °C, respectively. The following settings were used: sheath gas 40, auxiliary gas 5, and sweep gas 1, balanced AGC target. For positive mode ionization: spray voltage +4.5 kV, capillary voltage +40 V, tube voltage +70 V, skimmer voltage +20 V. For negative mode ionization: spray voltage −3.5 kV, capillary voltage −30 V, tube voltage −70 V, skimmer voltage −18 V.

Lipids were analysed using a Dionex UltiMate 3000 HPLC system (Thermo Fisher Scientific, Hemel Hempstead, UK). The method was modified based on a method previously described [50]. Lipid extracts were injected (10 µL, 6 °C) onto a reverse phase ACE Excel 2 SuperC18 column (50 × 2.1 mm; 2 μm particle size; Advanced Chromatography Technologies, Aberdeen, UK) held at 50 °C. Mobile phases consisted of A (60% water, 0.1% ammonium acetate, 40% acetonitrile) and B (10% water, 0.1% ammonium acetate, 10% acetonitrile, 80% isopropanol). The starting mobile phase was 30% B at a flow rate of 400 µL/min, and increased to 35% B by 1 min, then to 100% B by 7 min. The flow rate was then increased to 500 µL/min by 11 min. The proportion of mobile phase B decreased to 20% by 12 min, equilibrating for 3 min. Mass spectra of eluted extracts were collected in a Q-Exactive Plus hybrid quadrupole-Orbitrap mass spectrometer (Thermo Fisher Scientific, Hemel Hempstead, UK) fitted with a heated ESI (HESI) source. The polarity of the MS ionisation source was rapidly switched between positive and negative electrospray ionisation (ESI) modes. Full-scan data were acquired on the *m*/*z* 150–1500 range at 70,000 resolution. The probe and capillary temperature were maintained at 412.5 and 256.25 °C, respectively. The following settings were used: sheath gas 47.5, auxiliary gas 11.25, and sweep gas 2.25, AGC target 3 × 10^6^. The spray voltage was set to +4.0 kV or –4.0 kV. Data-dependent tandem MS/MS (ddMS^2^) spectra were produced on the 5 most intense ions at any one time at a resolution of 17,500, with AGC target 1 × 10^5^, injection time 50 ms, range 200–2000 *m*/*z*, normalised collision energy 30, isolation window 1.0 *m*/*z*, intensity threshold 1.6 × 10^5^, dynamic exclusion 10 s (Xcalibur 3.0.63 (Thermo Fisher Scientific, Hemel Hempstead, UK)).

### 3.8. Metabolite Identification

LC-MS raw metabolomics data from samples, blanks and QC samples were processed with XCMS for untargeted peak-picking [51], and peak matching and annotation of related peaks were carried out using mzMatch [52]. IDEOM with the default parameters was used for noise filtering and putative metabolite identification [53,54]. Briefly, retention time (RT) for the identification of authentic standards was 5%, RT for identification for calculated RT was 50%, and mass accuracy for mass identification was 3 ppm. LC-MS lipidomics raw datasets were processed using Progenesis QI (Nonlinear Dynamics, Waters, Newcastle upon Tyne, UK) which performed peak picking, peak matching and annotation using between-subject experimental design. Data were filtered using CV < 30% in QC samples and Lipid Maps with theoretical fragmentation were used for lipids putative identification. Lipid Search (Thermo Scientific, Hemel Hempstead, UK) was also used for lipid identification based on ddMS/MS, which uses an internal library and theoretical fragmentation with the addition of the ability to manually reject identifications that do not fit class trends in retention time (*t*-score), to process the data from the serum-starved to serum-replete validation experiment and the human tissue samples. Metabolites and lipids were identified with four levels of confidence; level 1 (L1) identification was based on matching the accurate masses, MS/MS fragmentation and retention times of the detected metabolite peaks with those of authentic standards which were co-analysed with the samples under identical experimental conditions, level 2 (L2) identification was based on matching the accurate masses and retention times (two orthogonal data) of the detected metabolite peaks with those of the authentic standards (e.g., the 268 metabolites run in the metabolomics sequence) or matching accurate masses and MS/MS spectra with compounds in a library when data were taken under the same acquisition parameters, level 3 (L3) identification was carried out when the predicted retention times or predicted MS/MS spectra or both were employed due to the lack of standards, and level 4 (L4) identification was based on unambiguously assigned molecular formulas, but insufficient evidence exists to propose possible structures [49]. The identification criteria were according to the metabolomics standards initiative and scale [29,55,56].

### 3.9. RNA and Transcriptomics Analysis

RNA was analysed using a Nanodrop (Thermo Fisher Scientific, Hemel Hempstead, UK) followed by a Tapestation instrument (Agilent Technologies, Santa Clara, CA, USA) to analyse the quantity and quality of RNA recovered by the mirVana^TM^ miRNA isolation kit and each of the metabolite/RNA dual extraction methods.

For transcriptomics analysis of serum starvation samples, the RNA was diluted to 600 ng/µL (concentrations determined by Nanodrop) and DNA was removed using the Turbo DNA-free kit (Invitrogen, Thermo Fisher Scientific, Hemel Hempstead, UK). The purified RNA was diluted to 100 ng/µL (concentrations determined by Tapestation) for a Transcriptomics Microarray. QC presented RIN of 9.7–10 (Bioanalyser; Agilent Technologies, Santa Clara, CA, USA). Whole-genome transcriptome analysis was conducted by hybridizing three biological samples of total RNA for treatment and four biological samples of total RNA for control to the Clariom^TM^ S arrays (#902926, Affymetrix, High Wycombe, Bucks, UK). All steps of sense cDNA synthesis, fragmentation and hybridization were performed according to the manufacturer’s protocol (GeneChip^®^ Scanner 3000 7G System, Affymetrix).

### 3.10. Statistical Methods

For LC-MS analysis, cell growth rate differed between serum-deplete and serum-replete samples, with 2.38 ± 0.57 million and 3.61 ± 0.40 million cells after 28 h, respectively, and therefore, the serum starvation result was normalised to the number of cells per sample (counted from a parallel culture). Student’s *t*-test and/or a one-factor ANOVA were computed to check for the significance of the metabolite/lipid fold changes [57,58]. The resulting *p*-values were then adjusted using the Benjamini–Hochberg false discovery rate to account for multiple testing problem (adjusted *p*-value < 0.05 was considered significant) [59]. SIMCA-P (v13, Umetrics, Umeå, Sweden) was used to conduct unsupervised principal component analysis (PCA) and orthogonal partial least squares-discriminant analysis (OPLS-DA) and give a list of variables important for the projection (VIP ≥ 1).

Microarray gene expression profile data were generated as CEL files and subjected to analysis by the Partek Genomics Suite 6.6 software (Partek, St. Louis, MO, USA). Quality Control (QC) metrics were checked by examining surface defects, hybridization, labelling, and a ratio of the 3′ probe set to the 5′ probe set (3′/5′ ratio) to provide the quality of the microarray data. The values were log^2^ transformed and quantile normalisation using the robust multi-array average (RMA). Filtering was performed to remove genes where expression did not differ from that in controls and genes where hybridization signals fell below the threshold for reliable detection using Partek Genomics Suite. The list of genes of interest comprised genes upregulated or downregulated by at least one-fold with Benjamini–Hochberg adjusted *p*-value < 0.05.

Multi-omics data integration of metabolites and transcripts was performed through pathway-based network analysis using Cytoscape (v.3.4.0) with MetScape 3 (v.3.1.3) plugin, applying a minimum fold change threshold of one and a *p*-value < 0.05 as all entered genes and metabolites were statistically significant, so no further filtering was required [2]. Network interactions were visualised using prefuse force directed layout.

## 4. Conclusions

A dual extraction method, sequential/mirVana, was developed and verified as a superior method for the extraction of both metabolites/lipids and RNA from a brain cancer cell line and primary brain tumour tissue and has potential for other applications. Simultaneous extraction of metabolites/lipids and RNA from the same sample set provides a consistent quality of the metabolome and RNA transcripts, and their biological relationship is adequately maintained and hence a better integrative metabolomics-transcriptomics can be obtained. The method was applied to distinguish between serum-starved and serum-replete cells on both the transcriptional and metabolic level after an acute period of perturbation and the data could be integrated to enhance the biological interpretation. This method will facilitate omics integration in a clinically relevant manner.

## Figures and Tables

**Figure 1 metabolites-11-00240-f001:**
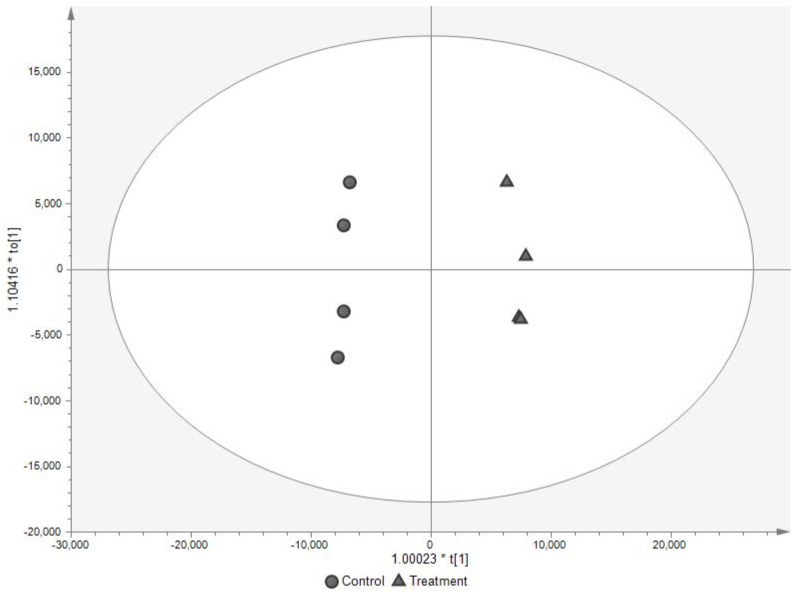
Statistical discrimination of treated (serum-starved) and control (serum-replete) BXD-1425 cells analysed by LC-MS; (**a**) metabolomics, (**b**) lipidomics. OPLS-DA scores plot. Four data points representing four replicates of each of treated (serum-starved, ▲) and control (serum-replete, ●) can be observed grouped together. (**a**) R^2^X 0.814, R^2^Y 0.995, Q^2^ 0.979; (**b**) R^2^X 0.880, R^2^Y 0.998, Q^2^ = 0.994.

**Figure 2 metabolites-11-00240-f002:**
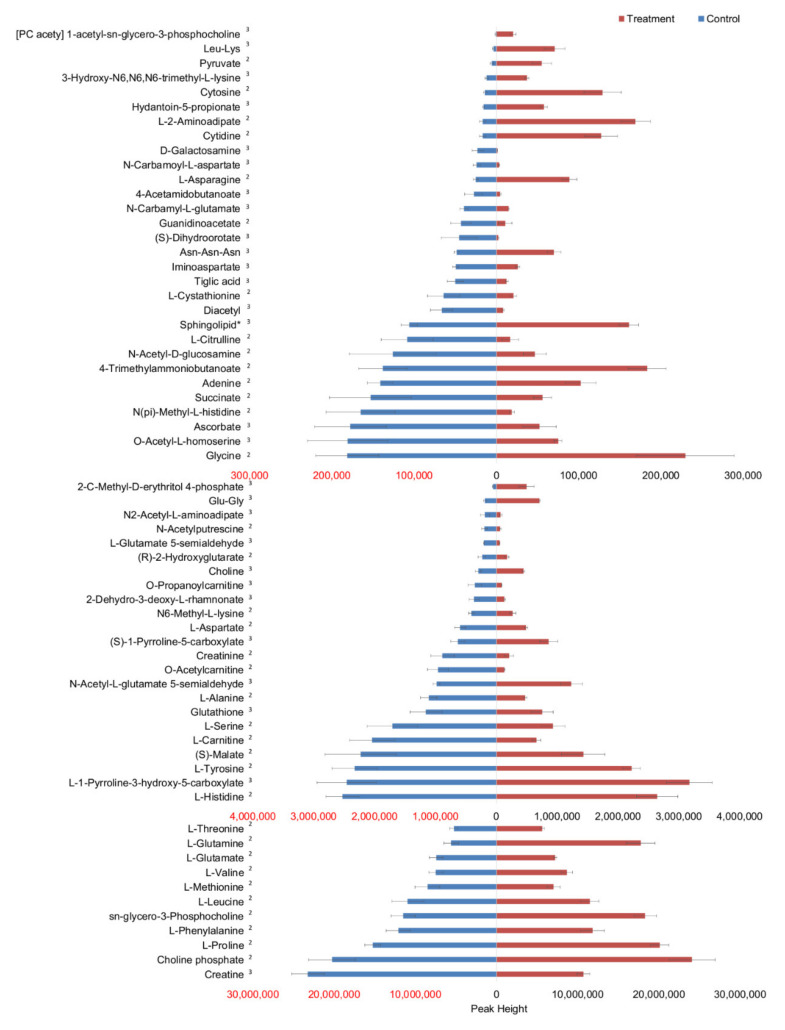
Differential accumulation of metabolites in treated (serum-starved) and control (serum-replete) cells. Differences in peak height were statistically significant for all metabolites displayed. Statistical significance was defined as a combination of *p* < 0.05 (in a univariate t-test with an FDR cut-off of 0.05) and VIP ≥ 1 (variable important for projection in a multivariate OPLS-DA test). The level of confidence of identification is given as superscripted numbers. * Sphingolipid is [SP hydroxy,hydroxy,methyl(10:2/2:0)] 6R-(8-hydroxydecyl)-2R-(hydroxymethyl)-piperidin-3R-ol.

**Figure 3 metabolites-11-00240-f003:**
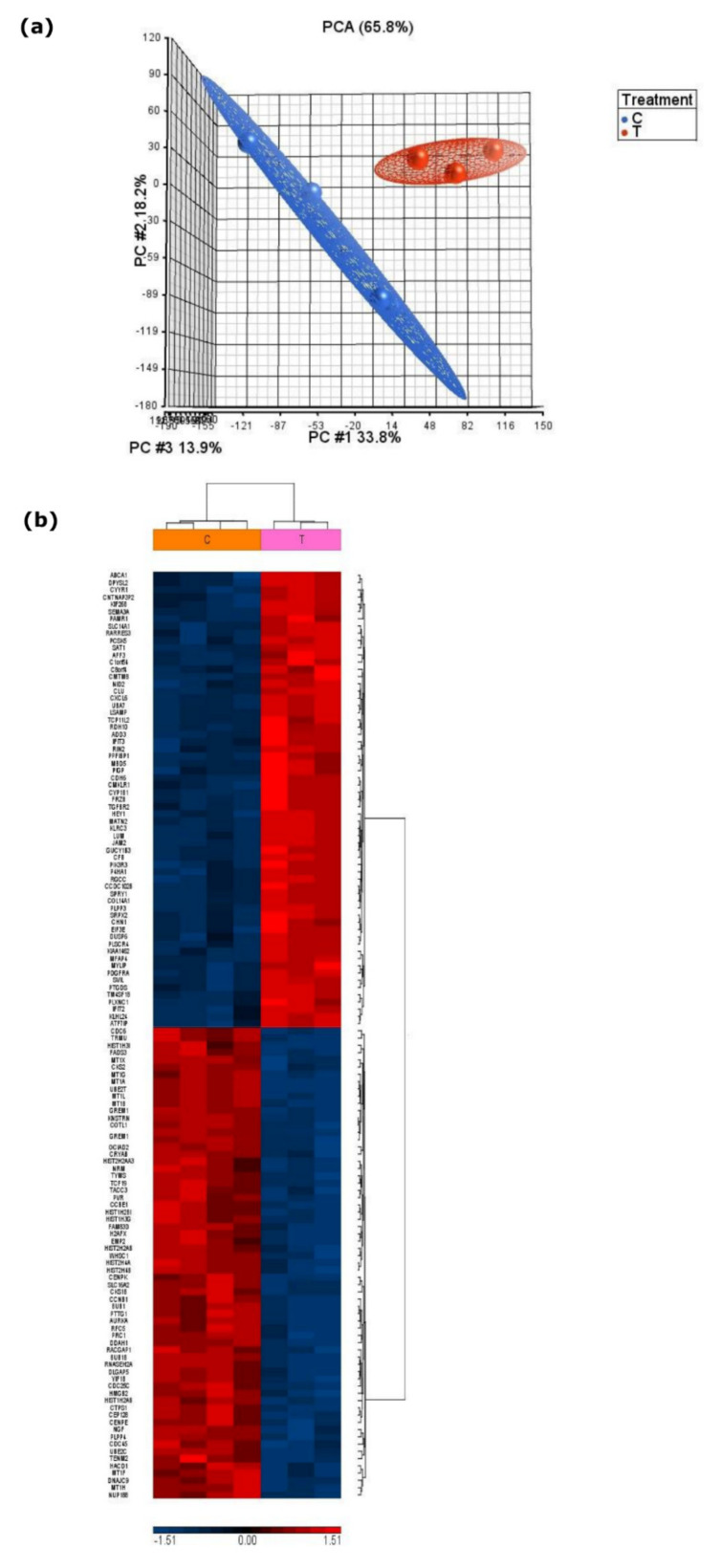
Statistical discrimination of treated (serum-starved) and control (serum-replete) BXD-1425 cells analysed by Affymetrix array. (**a**) PCA plot shows distinct clustering of treated (red) and control (blue) groups; (**b**) hierarchical clustering of the 128 differentially expressed genes. Intensity of colour is directly proportional to the difference in mean expression and ranges from blue (downregulated) to red (upregulated). C—control; T—treatment.

**Figure 4 metabolites-11-00240-f004:**
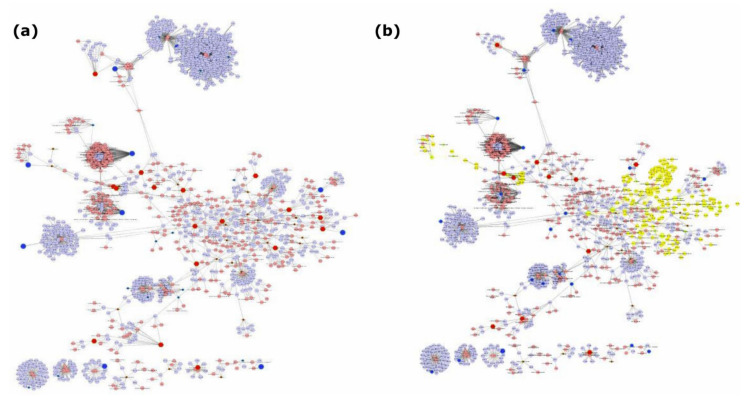
Integrated analysis of genes and metabolites extracted from treated (serum-starved) relative to control (serum-replete) BXD-1425 cells. (**a**) Compound reaction networks of the metabolites and genes were visualised using MetScape: metabolites (red) and genes (blue) are presented as nodes. (**b**) The metabolite–gene associated networks were mainly related to purine metabolism (yellow).

**Figure 5 metabolites-11-00240-f005:**
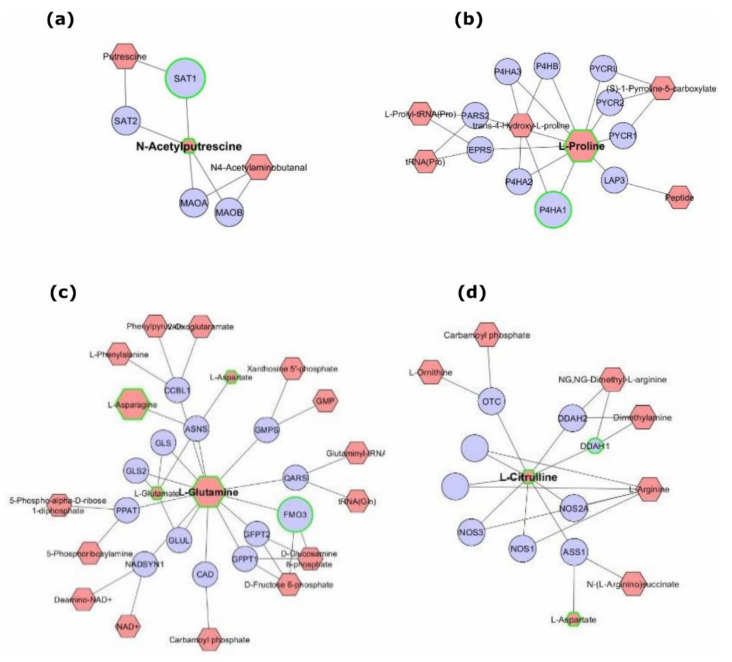
Integrated gene–metabolite interactions facilitated by dual-extraction from the same population of cells. The depicted networks reveal aberrant metabolites (red) associated with dysregulated genes (blue) between treated and control BXD-1425 cells (>1.0 fold). The integrative network was generated using a MetScape plugin for Cytoscape. Significantly altered genes or metabolites (green border) in serum-starved relative to serum-replete cells are represented as upregulated/downregulated or high/low abundance, respectively, by an increase or decrease in node size in comparison to other genes or metabolites. (**a**) Direct statistically significant edge between N-acetylputrescine and SAT1, (**b**) edge between L-proline and P4HA1, (**c**) edge between L-glutamine and FMO3, (**d**) edge between L-citrulline and DDAH1.

**Figure 6 metabolites-11-00240-f006:**
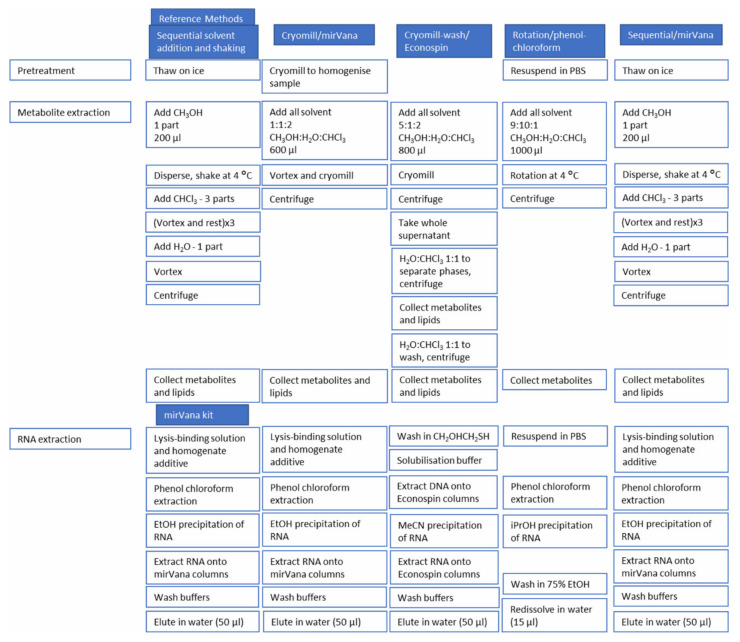
Schematic of metabolite/RNA dual extraction procedures and the single extraction reference methods.

**Table 1 metabolites-11-00240-t001:** A metabolomics comparison of five metabolite extraction methods using BXD-1425 cells.

Extraction Method	Sequential Solvent Addition and Shaking (Positive Control)	Cryomill/mirVana	Cryomill-Wash/Econospin	Rotation/Phenol-chloroform	Sequential/mirVana
*n*	3	2	4	3	4
Total ion count	4.95 × 10^7^ ± 1.53 × 10^6^	5.58 × 10^7^ ± 2.17 × 10^6^	5.80 × 10^7^ ± 3.66 × 10^6^ *	5.35 × 10^7^ ± 8.97 × 10^6^	5.01 × 10^7^ ± 5.50 × 10^6^
Number of ions (unfiltered)	2025 ± 2	2022 ± 1	2024 ± 2	2013 ± 5	2025 ± 3
Number of metabolites	193	193	193	193	193
Significantly altered metabolites *	-	31 (22 ↑, 9 ↓)	0	1 ↓	0
Practical comments		-Cryo-milling can damage tubes and leak solvents –Quick protocol which can use any RNA extraction kit.	-Cryo-milling can damage tubes and leak solvents. -Long protocol involving washing steps purifies metabolites but is time-consuming. Cheap, but time-consuming RNA extraction as uses self-made buffers.	Metabolite extraction protocol is simple but is not biphasic so cannot perform lipidomics on a C18 column.	Easy to perform, possibility to extract microRNA separately. Longest solvent evaporation time.

* indicates a significant difference (*p* < 0.05) from the positive control. ↑ indicates number of metabolites increased in abundance and ↓ indicates number of metabolites decreased in abundance compared to the positive control. All other comparisons were non-significant. Sequential solvent addition and shaking was the extraction of metabolites alone; in the other methods, metabolites were extracted from dual extraction methods. Metabolites were analysed by LC-MS and were identified either by retention time and exact mass match with authentic standards (level 2—55 identifications), or by using the retention time prediction model, IDEOM (putative identification level 3—138 identifications).

**Table 2 metabolites-11-00240-t002:** A lipidomics comparison of four lipid extraction methods using BXD-1425 cells.

Extraction Method	Sequential Solvent Addition and Shaking (Positive Control)	Cryomill/mirVana	Cryomill-Wash/Econospin	Sequential/mirVana
*n*	3	2	4	4
Total ion count—negative	3.41 × 10^9^ ± 1.44 × 10^8^	3.36 × 10^9^ ± 1.80 × 10^8^	3.74 × 10^9^ ± 3.35 × 10^8^	3.45 × 10^9^ ± 3.52 × 10^8^
Total ion count—positive	7.12 × 10^9^ ± 2.33 × 10^8^	6.48 × 10^9^ ± 4.33 × 10^8^	7.03 × 10^9^ ± 6.04 × 10^8^	7.40 × 10^9^ ± 7.28 × 10^8^
Number of putative lipids	3390	3428	3506	3427
Significantly altered lipids *	-	356 (256 ↑, 100 ↓)	618 (394 ↑, 224 ↓)	314 (244 ↑, 70 ↓)

* indicates a significant difference (*p* < 0.05) from the positive control. ↑ indicates number of lipids increased in abundance and ↓ indicates number of lipids decreased in abundance compared to the positive control. All other comparisons were non-significant. Rotation/phenol-chloroform did not extract lipids. Sequential solvent addition and shaking was the extraction of lipids alone; in the other methods lipids were extracted from dual extraction methods. Lipids were analysed by LC-MS/MS and were identified by exact mass match and MS2 match with a library and fragmentation predictor, respectively (putative identification level 3).

**Table 3 metabolites-11-00240-t003:** Yield and quality of RNA extracted from BXD-1425 cells.

Extraction Method	mirVana Kit (Positive Control)	Cryomill/mirVana	Cryomill-Wash/Econospin	Rotation/Phenol-Chloroform	Sequential/mirVana
Concentration of RNA (ng/µL)	1300 ± 21	910 ± 26 *	514 ± 29 *	1331 ± 249	887 ± 131 *
Mass of RNA (µg)	65.0 ± 1.3	45.5 ± 1.8 *	25.7 ± 2.0 *	20.0 ± 4.6 *	44.4 ± 7.6 *
Relative % of RNA recovered	100	70*	40 *	31 *	68 *
260/280 ratio	2.10 ±0.01	2.05 ±0.01	2.16 ±0.01 *	2.11 ±0.01	2.06 ±0.01 *
260/230 ratio	1.80±0.47 ^a^	2.03 ±0.15	1.97 ±0.02	2.09 ±0.02	2.05 ±0.07
28S/18S (Area)	2.9 ± 0.1	2.6 ± 0.1	2.7 ± 0.1	1.4 ± 0.1 *	2.2 ± 0.7
RIN	9.4 ± 0.1	7.9 ± 0.5	9.1 ± 0.2	7.6 ± 0.3 *	8.9 ± 0.1 *

* indicates a significant difference (*p* < 0.05) from the positive control. All other comparisons were non-significant. The mirVana^TM^ miRNA isolation kit was the extraction of RNA alone; in other methods, the RNA was extracted from dual extraction methods. The results of Nanodrop and Agilent Tapestation measurements are shown. The RNA solution from the extraction procedure was analysed by Nanodrop and subsequently, 1 µL of RNA extracted via each method (and diluted to an approximate concentration of 400 ng/µL) was applied to an RNA nanochip and run alongside the electronic ladder. Numbers are means from two, three or four independent extractions per method. ^a^ One anomalous result of 1.13 has caused this large standard deviation.

**Table 4 metabolites-11-00240-t004:** Differentially expressed genes mapping onto metabolic pathways.

Gene	Metabolic Pathway	*p*-Value	Fold-Change
ATP1A1	Purine metabolism	1.69 × 10^5^	1.37693
AURKA	Glycosphingolipid metabolism	6.37 × 10^5^	−4.80815
BUB1	Glycosphingolipid metabolism	1.39 × 10^4^	−4.57951
BUB1B	Glycosphingolipid metabolism	1.64 × 10^5^	−3.05730
CDC25C	Glycosphingolipid metabolism	6.00 × 10^5^	−2.45823
CTPS1	Pyrimidine metabolism	2.03 × 10^4^	−2.73359
CYP1B1	Arachidonic acid metabolism	2.18 × 10^5^	4.91775
DDAH1	Arginine, proline, glutamate, aspartate and asparagine metabolism	4.29 × 10^5^	−2.61779
DHFR	Folate metabolism	1.76 × 10^4^	−1.89205
DUSP6	Glycosphingolipid metabolism	1.98 × 10^4^	3.58612
FADS3	Omega-6 fatty acid metabolism	2.04 × 10^4^	−2.85552
FMO3	Amino sugars metabolism	2.41 × 10^4^	1.80707
GUCY1B3	Purine metabolism	1.36 × 10^5^	10.77960
NDST3	Proteoglycan metabolism	6.77 × 10^5^	1.60250
NDUFB2	Methionine metabolism	1.27 × 10^4^	−1.47355
P4HA1	Arginine, proline, glutamate, aspartate and asparagine metabolism	1.73 × 10^4^	3.91765
PLPP3	Glycerophospholipid metabolism	7.39 × 10^6^	2.52901
PTGDS	Prostaglandin formation from arachidonate	1.55 × 10^4^	2.43688
RFC5	Purine metabolism	2.00 × 10^4^	−2.96426
SAT1	Arginine, proline, glutamate, aspartate and asparagine metabolism	6.22 × 10^6^	6.06009
TGFBR2	Glycosphingolipid metabolism	1.11 × 10^4^	2.51642
TRMU	Methionine metabolism	2.30 × 10^4^	−2.36320
TYMS	Folate metabolism	1.71 × 10^4^	−3.64677
UBE2C	Purine metabolism	2.10 × 10^4^	−6.48322
UBE2T	Purine metabolism	3.33 × 10^6^	−3.90861
WHSC1	Lysine metabolism	7.95 × 10^5^	−2.65510

Fold changes are relative to serum-replete control cells.

## Data Availability

Data available in a publicly accessible repository. The data presented in this study are openly available in the Nottingham Research Data Management Repository at https://rdmc.nottingham.ac.uk (accessed on 14 April 2021); reference number [doi: 10.17639/nott.7113].

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
