# Peer review of "Integrated Metabolomics and Transcriptomics Using an Optimised Dual Extraction Process to Study Human Brain Cancer Cells and Tissues"

_metabolites, 2021, doi:10.3390/metabo11040240_

Round 1

Reviewer 1 Report

The manuscript of Woodward et al.,  described a dual extraction method  (sequential/mirVana) for the extraction of metabolites/lipids and RNA from a brain cancer cell line and primary brain tumor tissues. The authors showed that this method is superior than other methods in the extractions and can be applied to distinguish between serum-starved and serum-replete cells. The paper is interesting and beneficial to this field.

The topic is original since it compares different extraction procedures for metabolites/ lipid and/or RNA and how the extraction method affect efficiency of downstream procedure such as omics, metabolomics, and transcriptomics. In addition, the authors showed that the best extraction method (sequential/mirVana) can be used to distinguish between serum- starved and serum-replete cells after an acute period of perturbation.   This manuscript can be beneficial in the field of metabolomics and transcriptomics since it gives a solution for some troubleshootings appeared with other researchers in the field.   The paper describes an accurate methodology and procedure to improve the efficiency of integrated metabolomics and transcriptomics in the human brain cancer cells and tissues.   Other researchers in the field can utilize the data in this manuscript to improve the sample processing and analysis.   The follow of paper is good, and it is well written Minor English editing is required.

Missing points:

No statistic results are shown by the authors, especially table 1,2, and 3. It is important to show if there is a significant difference between the different extraction procedures.

Reviewer 2 Report

Review: Integrated Metabolomics and transcriptomic using an optimized dual extraction process to study human brain cancer cells and tissue

  • Abstract: Summarizes findings of the study well. Authors discuss dual-extraction process in ependimomas.
  • Introduction is very detailed, but at the same time quite lengthy. I suggest that it may be shortened.
  • Results/Discussion section presents informative metabolite and lipid extraction methods, therefore text may be reduced.
  • Page 9, line 318, I did not understand why it was able to detect intracellular metabolic changes.
  • Figure 5 should say in explanation what is a,b,c,d because it is confusing
  • Article repeats certain things, which can be avoided in order to reduce volume
  • In general, data are good even though only one patient is mentioned. It would be good to see more explanation of why certain metabolite-RNA data were found and what clinical application it may have. For example sphingolipids, but also other metabolites.
